# Effects of Co-Fermentation of *Lactiplantibacillus plantarum* and *Saccharomyces cerevisiae* on Digestive and Quality Properties of Steamed Bread

**DOI:** 10.3390/foods12183333

**Published:** 2023-09-06

**Authors:** Yan Liu, Muhammad Danial, Linlin Liu, Faizan Ahmed Sadiq, Xiaorong Wei, Guohua Zhang

**Affiliations:** 1School of Life Science, Shanxi University, Taiyuan 030006, China; 202123118023@email.sxu.edu.cn (Y.L.); mrdanial.pk@gmail.com (M.D.); 19834427301@163.com (X.W.); 2State Key Laboratory of Food Nutrition and Safety, Food Biotechnology Engineering Research Center of Ministry of Education, Tianjin University of Science and Technology, Tianjin 300457, China; linlin150314@163.com; 3Technology & Food Sciences Unit, Flanders Research Institute for Agriculture, Fisheries and Food (ILVO), 9090 Melle, Belgium; faizanahmed.sadiq@ilvo.vlaanderen.be

**Keywords:** *Lactiplantibacillus plantarum* (LP-GM4), *Saccharomyces cerevisiae* (yeast), starch digestion, protein digestion, steamed bread

## Abstract

The leavening of wheat-based steamed bread is carried out either with a pure yeast culture or with traditional starter cultures containing both lactic acid bacteria and yeast/mold. The use of variable starter cultures significantly affects steamed bread’s quality attributes, including nutritional profile. In this paper, differences in physicochemical properties, the type of digested starch, the production of free amino acids, and the specific volume of steamed bread under three fermentation methods (blank, yeast, and LP-GM4-yeast) were compared. The digestion characteristics (protein and starch hydrolysis) of steamed bread produced by using either yeast alone or a combination of *Lactiplantibacillus plantrum* and yeast (LP-GM4-yeast) were analyzed by an in vitro simulated digestion technique. It was found that the specific volume of steamed bread fermented by LP-GM4-yeast co-culture was increased by about 32%, the proportion of resistant starch was significantly increased (more than double), and soluble protein with molecular weight of 30–40 kDa was significantly increased. The results of this study showed that steamed bread produced by LP-GM4-yeast co-culture is more beneficial to human health than that by single culture.

## 1. Introduction

Steamed bread is a traditional fermented food that has been consumed in China for hundreds of years as a staple food [1], accounting for 40% of the total flour used in China [2]. Steamed bread is traditionally fermented at home or in small bakeries using sourdough or yeast as a starter. Sourdough is the primary leavening agent for its production [3]. Sourdough starter has the function of improving product sensory quality, flavor, texture, increasing nutritional value and extending shelf life [4]. Sourdough is a mixture of flour and water containing a diversity of lactic acid bacteria (LAB) and yeast/fungal species [5]. It was used for thousands of years before the advent of industrial yeast, and is still widely favored today [6]. According to past studies, sourdough starters are typically composed of 50 LAB species (mainly *Lactobacillus*) and more than 20 yeast species (mainly *Saccharomyces* and *Candida*) [7,8,9]. The most dominant LAB species of a typical sourdough microbial niche are *Lactiplantibacillus plantrum*, *Levilactobacillus brevis*, and *Fructilactobacillus sanfranciscensis* [10]. The most dominant yeast species are *Saccharomyces cerevisiae*, *Candida humilis*, and *Torulaspora delbrueckii* [11]. Myriad studies have reported sourdough microbiota’s influence on its aroma profile [12,13] and nutritional value [14]. It is pivotal to investigate further nutritional value of steamed bread prepared by adding different starters as leavening agents. *Lactiplantibacillus plantrum* (*L. plantarum*) is frequently employed as a starter during fermentation of steamed bread. This bacterial fermentation facilitates acidification and enzymatic hydrolysis, which increase the solubility of proteins and hence improve the elasticity and ductility of the dough [15]. On the contrary, *Saccharomyces cerevisiae* (*S. cerevisiae*) functions as a leavening agent to eliminate the starch and protein through hydrolysis [16,17,18]. The in vitro digestion technique is used by scientists to study starch and protein content and biologically active substances [19]. In recent years, in vitro digestion has become a more practical and reliable method to study various metabolic processes outside the human body using simulated conditions of metabolism and enzymatic reactions [20]. This technique generally includes simulated digestion compartments and secretory enzymes that are produced during oral, gastric, and intestinal digestion.

Empirical, traditional, and in vitro scientific findings all evidence that sourdough’s long fermentation process increases bread digestibility. Moreover, bread digestibility is determined by a variety of parameters, including appetite perception, satiety, gastrointestinal symptoms after the consumption of bread [21,22,23], and the starch and protein bioavailability. Protein degradation as a result of sourdough fermentation increases the potentially health-promoting peptides and short-branched-chain amino acids. These compounds have been found to support the regulation of insulinemic response and reduce the risk of related diseases such as cardiovascular problems and diabetes [24]. Inoculation of particular LAB strains in sourdough improves the proteolysis process and successfully doubles the content of amino acids including lysine, isoleucine, leucine, and histidine [25].

Starch is the major component of cereals, and starch digestibility during sourdough fermentation has been the subject of scientific research for years [26]. The research suggests that sourdough’s action on starch digestion may rely on the lactic and acetic acid produced by sourdough-associated species. Sourdough fermentation significantly delays starch digestion, which can promote the formation of indigestible polysaccharides that leave the small intestine with grain fibers, which are then fermented by colonic microbiota [27].

Compared with the industrial steamed bread fermented by single yeast, the traditional steamed bread with natural fermentation has a significant advantage in flavor. However, at present, there are still many technical difficulties in simulating the complex multi-strain cooperative fermentation under the natural state, which seriously hinders the industrialization process of traditional steamed bread. Therefore, we chose to carry out work on the co-culture of two strains with relatively simple composition. This study conducted in vitro digestion research on steamed bread produced by different fermentation methods, and carried out professional sensory evaluation on them, explained the differences between individual culture and co-culture from the physiological and sensory levels, and analyzed the causes of differences. It not only lays a foundation for the simulation of a more complex multi-strain cooperative system in the future but also provides data support for the industrial production of multi-strain fermented steamed bread.

## 2. Materials and Methods

### 2.1. Materials

In this study, the LAB species (*L. plantarum* strain (LP-Gm4)) was used, preserved in the School of Life Sciences, Shanxi University, Taiyuan, and the MRS medium provided by Beijing Aobo Star Biotechnology Co., Ltd. was used to grow the bacterial and yeast species. The Wudeli flour was provided by Wudeli Flour Group, while the active dry yeast was provided by Lesaffre Management Co., Ltd., Shanghai, China. The α-amylase enzyme and pepsin were provided by Maclean Biochemical Technology Co., Ltd., Shanghai, China.

### 2.2. Preparation of Different Types of Steamed Bread

To prepare the yeast-group steamed bread, add 0.5 g active dry yeast to 50 g flour, add 25 mL deionized water, stir in a flour mixer for 10 min, fold and knead the dough by hand 10 times, then cut the dough into 10 g portions and knead by hand 15–20 times until the dough is moistened and formed. Leave in a box with 85% humidity and 30 °C temperature for 30 min to rise, and then steam in a pot for 20 min [28]. The cultured *L. plantarum* bacterial solution was centrifuged at 5000 rpm/min for 10 min, and the obtained bacterial slurry was mixed with 50 g of flour, 25 mL of water, and 0.5 g of active dry yeast. The preparation method for the LP-GM4-yeast co-culture group was the same as the yeast group. For the blank group, steamed bread was prepared using an equal proportion of flour and water to that used for the yeast group.

### 2.3. pH and TTA

pH and TTA (total titratable acidity) values were measured according to the national standard GB/T 21118-2007 [29]. [M1] To measure the pH, 10 g of the sample was homogenized into 100 mL of (sterile) distilled water. Acidity was titrated with 0.1 mol/L NaOH to obtain a pH of 8.2 [12]. TTA is expressed in milliliters of 0.1 mol/L NaOH, the unit is grams per kilogram (g/kg), and the results are accurate to one decimal place. All the experiments were performed in triplicate. TTA value was calculated according to the following formula:(1)$$X=\frac{0.1\times (V_{1}{-V}_{2})\times 0.090}{10}\times 1000,$$

V_1_—sample solution consumes NaOH standard solution volume/mL.V_2_—blank consumption NaOH standard solution volume/mL.0.090—conversion factor for lactic acid.

### 2.4. Determination of Specific Volume

According to the GB/T 21118-2007 appendix [29], the millet seed displacement principle was used to measure the specific volume of steamed bread. The final results were calculated according to the following formula:(2)$$\lambda =\frac{V}{m},$$

λ—the specific volume of steamed bread in mL/g.V—the volume of steamed bread volume in mL; m is the mass of steamed bread (g)

### 2.5. Free Glucose Mass Fraction and Total Starch Determination

Free glucose (FG) was determined according to the method previously described by Niu et al., 2020 [30]. A 1 g sample and phosphate buffer (0.2 mol/L, pH 5.2) were thoroughly mixed and heated for 20 min at 95 °C. After incubation at room temperature (37 °C), the gelatinized starch sample was centrifuged at 5000 r/min for 10 min [30]. The glucose content in the supernatant was determined using the DNS technique, and the mass fraction of FG in starch was calculated by the following formula:(3)$$\mathrm{FG}=\frac{\mathrm{Glucose}\;\mathrm{contnet}\;\mathrm{in}\;\mathrm{supernatant}}{\mathrm{quality}\;\mathrm{of}\;\mathrm{starch}\;\mathrm{sample}}\times 100\%,$$

The 1 g crushed sample was accurately weighed and transferred into a 100 mL triangular/conical flask, followed by 15 mL distilled water and 10 mL of HCl being added at a concentration of 6 mol/L. The solution was then hydrolyzed for 30 min using a heating medium (boiling water). After cooling the hydrolysate in the triangle/conical flask, it was neutralized with 6 mol/L NaOH until it turned reddish. Before the filtration process, distilled water was added to obtain a constant volume of 100 mL. The resulting 10 mL filtrate was transferred into a volumetric flask after mixing to determine the total starch mass fraction.
(4)$$\mathrm{TS}=\frac{(\mathrm{Glucose}\;\mathrm{content}\;\mathrm{in}\;\mathrm{supernatant}\;-\;\mathrm{FG})\times 0.9}{\mathrm{quality}\;\mathrm{of}\;\mathrm{starch}\;\mathrm{sample}}\times 100\%,$$

### 2.6. Determination of Digested Starch Types

The in vitro starch digestibility was analyzed using Chi et al.’s technique [31]. The calculation formulas are given below:(5)$$\mathrm{RDS}=\frac{{(G}_{20\min}-\mathrm{FG})\times 0.9}{\mathrm{quality}\;\mathrm{of}\;\mathrm{sample}}\times 100\%,$$
(6)$$\mathrm{SDS}=\frac{(G_{120\min}{-G}_{20\min})\times 0.9}{\mathrm{quality}\;\mathrm{of}\;\mathrm{sample}}\times 100\%,$$
(7)$$\mathrm{RS}=\frac{\mathrm{TS}-\mathrm{RDS}-\mathrm{SDS}}{\mathrm{quality}\;\mathrm{of}\;\mathrm{sample}}\times 100\%$$
where G_20_ and G_120_ are the glucose content after 20 and 120 min digestion, respectively; TS is total starch content; RDS is rapidly digested starch; SDS is slowly digested starch; and RS is resistant starch.

### 2.7. In Vitro Gastrointestinal Digestion of Steamed Bread Prepared with Different Starter Cultures

A sample of 3 g steamed bread was thoroughly mixed with 5 mL of simulated saliva fluid (SSF) and 75 U/mL of α-amylase, adjusted to pH 7.0, and then digested in a water bath at room temperature for 2 min. Similarly, 2000 U/mL of pepsin was added to the same volume of steamed bread and simulated gastric fluid (SGF), the pH adjusted to 1.5, and then digested for 2 h. Following the gastric digestion, the same simulated intestinal fluid (SIF) content was mixed with 200 U/mL of pancreatic amylase to achieve a pH of 7.2 and digested at room temperature for 2 h. Simulated in vitro gastrointestinal digestion was performed using artificial gastric enzymes (Table 1), which were prepared using the method described by Brodkorb et al. [32].

### 2.8. Amino Acid Analysis

The free amino acids produced during the digestion of steamed bread were analyzed using an auto amino acid analyzer. The samples were pretreated according to the national standard GB 5009.124-2016 [33]. Samples of 5 g steamed bread were crushed with a tissue grinder, and 1 g of each crushed sample was accurately weighed and placed in a hydrolysis tube. Add 15 mL hydrochloric acid solution to the hydrolysis tube, and continue to add 3 to 4 drops of phenol to the hydrolysis tube. Put the hydrolysis tube into the refrigerant, freeze it for 3 to 5 min, vacuum it, then fill it with nitrogen, repeat vacuum, fill it with nitrogen three times, and seal it when it is filled with nitrogen. The sealed hydrolysis tube was placed in an electric blast incubator at 110 ± 1 °C, and after hydrolysis for a certain period of time, it was removed and cooled to room temperature. Open the hydrolysis tube, filter the hydrolysate into a 25 mL volumetric bottle, rinse the hydrolysis tube with a small amount of water several times, transfer the water lotion into the same 25 mL volumetric bottle, adjust the volume with water to scale, and shake well. Accurately absorb 0.5 mL filtrate and transfer it into a 10 mL test tube, use concentrator to reduce pressure and dry under 40–50 °C heating environment. After drying, the residue is dissolved with water. Then dry under pressure, and finally leave to dry completely.

Citric acid buffer solution (0.5 mL), pH 2.2, was added into the dried test tube to shake and dissolve. After vortex mixing, the solution was absorbed and passed through the film for instrument determination. The mixed amino acid standard solution and sample determination solution were injected into the amino acid analyzer, and the amino acid concentration in the sample determination solution was calculated by the peak area of the external standard method.

### 2.9. Sensory Evaluation of Steamed Bread

The sensory evaluation of steamed bread was carried out according to the method in GB/T 35991-2018 [34]. A steamed bread sensory evaluation team composed of 15 professionals conducted sensory evaluation on three kinds of steamed bread in terms of five qualities—appearance, texture, flavor, elasticity, and internal structure—according to the scoring rules in Table 2.

### 2.10. Statistical Analysis

For data analysis, a one-way analysis of variance (ANOVA) was used. To present all data, the mean ± standard deviation was used, and *p* values less than 0.05 were regarded as statistically significant. All the experiments were performed in triplicate.

## 3. Results and Discussion

### 3.1. pH and TTA

LAB can metabolize carbohydrates in steamed bread to produce lactic acid, propionic acid, and some other acids that decrease the pH of the dough during fermentation. In the current study, before fermentation, the pH of the three experimental groups (blank, yeast, and LP-GM4-yeast group) was in a similar range (of around 6.10) with no significant variation (Figure 1). During the fermentation, the pH values of all three groups showed a downward trend, whereas the TTA value displayed an upward trend (Figure 2). Furthermore, the pH and TTA values of the yeast group (*S. cerevisiae*) showed no significant difference compared with the blank group. However, the pH and TTA values for the LP-GM4-yeast group decreased significantly, indicating that the synergistic fermentation of LP-GM4 yeast resulted in the production of additional organic acids. Our findings are in agreement with Teleky et al. 2020, who reported that the co-fermentation of LAB and yeast species produced higher lactic acid content than the yeast alone [35]. Yeast generally produces carbon dioxide and ethanol during sourdough fermentation. The metabolic activity of sourdough depends on LAB and yeast interaction, including the bioconversion of carbohydrates into organic acids and other chemical compounds [36].

### 3.2. Comparison of Starch Types

The comparison of rapidly digestible starch (RDS), slowly digestible starch (SDS), and resistant starch (RS) in steamed bread produced with yeast and the LP-GM4 yeast is presented in Figure 3. Compared to the blank group, the RDS and SDS contents in the yeast group were increased by two times, whereas the content of RS was significantly decreased (Figure 3). Furthermore, the RDS content of the LP-GM4-yeast steamed bread was increased 1.5 times approximately compared to the blank group, while the SDS content of the blank group steamed bread was reduced to 50%. Comparing the LP-GM4-yeast steamed bread with active dry yeast, the RS of the LP-GM4-yeast group was greatly increased (<2 times). However, RDS and SDS were relatively reduced, attributed to synergistic fermentation of LP-GM4 yeast.

Sourdough microbial fermentation significantly affects the reduction of starch and protein digestibility, e.g., the production of organic acids by LAB during the fermentation decreases starch digestion [8,37]. Our study revealed that during the fermentation of steamed bread, co-culture of LP-GM4 and yeast produced more resistant starch (RS), resulting in a decrease in hydrolysis rate and glycemic index. Earlier, De Vuyst et al. in 2009 found that organic acid production by LAB during the fermentation decreased starch digestion [8,37]. In the current study, when *L. plantarum* (LP-GM4) was added to yeast, it produced organic acids (lactic acid and acetic acid) during fermentation, which increased the RS content in the steamed bread. Acetic acid appears to be associated with a delay in gastric emptying [38], whereas lactic acid induces interactions between starch and gluten during dough baking and hence reduces starch availability [39]. The addition of sourdough or LAB as starter in steamed bread can increase the RS and SDS content and decrease the RDS content [40]. Earlier studies have reported that during fermentation, LAB species produce more organic acids (primarily lactic acid and acetic acids) that can lower the glycemic index and reduce the starch bioavailability [41].

### 3.3. Amino Acid Analysis

In steamed bread prepared using different starters (blank, yeast, and LP-GM4-yeast groups), a total of 17 amino acids were identified, including 7 essential amino acids, 2 semi-essential amino acids (His, Arg), and 8 non-essential amino acids (Table 3) The contents of Thr, Val, Met, Lys, and other amino acids increased in the LP-GM4-yeast group, and the EAA (essential amino acids)/NEAA (non-essential amino acids) and EAA/TAA levels of the LP-GM4-yeast group were also greater than the blank and yeast groups. The content of amino acids such as lysine, phenylalanine, methionine, isoleucine, and others was improved in the LP-GM4-yeast group. Lysine, also known as wheat flour’s first limiting amino acid, regulates protein absorption in the human body [42], and the lysine content of the LP-GM4-yeast group steamed bread was raised. Isoleucine is an essential amino acid that can contribute to the body’s glucose and fat metabolism, and it also promotes protein synthesis and hormone production in the human body [43]. Methionine plays a significant function in the human body, for instance, participation in DNA and protein synthesis is one of its physiological and biochemical activities, and metabolite S-adenosylmethionine of methionine can indirectly regulate various metabolic processes [44]. The steamed bread produced by co-fermentation of LP-GM4 yeast may be beneficial to the absorption of protein and promote protein synthesis and hormone production in the human body.

### 3.4. Changes in Specific Volume of Steamed Bread

The specific volume of the steamed bread obtained by synergistic fermentation of LP-GM4 yeast was significantly increased compared to the blank and yeast group steamed bread by 175% and 32%, respectively (Figure 4). Jarosz et al., 2014 demonstrated that lactic acid produced by LAB during fermentation induced glucose suppression, unavailability of glucose enabled yeast cells to metabolize a large variety of carbohydrates that enhanced the absorption of limiting nutrients, and prolonged yeast chronological lifetime [45]. We found similar results: the addition of LP-GM4 produced higher lactic acid content and quickly reduced the pH of the dough to 3.5, while the low-pH environment promoted the yeast (*S. cerevisiae*) growth vigorously. *S. cerevisiae* is a leavening agent that reduces simple carbohydrates (such as glucose) to CO_2_ and ethanol, which changes the physical properties of the dough [46]. The formation of CO_2_ gas by yeast is retained in the dough by a complex viscoelastic wheat gluten network [47], and leads to dough leavening, and thus it is indicative of the final bread volume.

### 3.5. Sensory Evaluation

A review panel of food specialists evaluated the appearance, internal structure, flavor, elasticity, and taste of the steamed bread (Figure 5) produced using three different starter agents (blank, yeast, and LP-GM4 yeast). The yeast group steamed bread was shown to have the lowest shelf life and also obtained low scores in terms of sensory properties according to the scoring criteria of Table 3. Notably, the LP-GM4-yeast group steamed bread was given the highest score of 87 points. The results showed that shelf life and sensory attributes of the LP-GM4-yeast group were improved significantly. This group was also given the highest points in terms of taste and aroma, which may be attributable to the synthesis of certain alcohols, acids, and esters, including lactic acid, acetic acid, propionic acid, ethanol, n-propanol, isopropanol, ethyl acetate, and so on. Additionally, the degradation of protein generates certain amino acids that can be employed as precursors for synthesizing flavor substances that enhance the steamed bread’s flavor. After an assessment of all three samples (blank, yeast and LP-GM4-yeast groups), it was found that steamed bread of the nLP-GM4-yeast group, which was fermented with the co-fermentation of *L. plantarum* and *S. cerevisiae*, was the most appreciated among other groups in terms of sensory attributes. Edeghor et al., 2016 also reached similar results: their research demonstrated that synergistic interaction between LAB and baker’s yeast (*S. cerevisiae*) enhances the sensory characteristics of bread and also prolongs shelf life [48].

### 3.6. In Vitro Protein and Starch Digestion

#### 3.6.1. Starch Digestion

After the simulated oral, stomach, and intestinal digestion, steamed bread is converted into maltose by the function of the α-amylase enzyme, which is later converted into glucose and utilized as an energy source in small intestinal epithelial cells. The amount of glucose produced in the steamed bread co-fermented with LP-GM4 yeast during the in vitro digestion process was less than that produced by the yeast group (Figure 6) and gradually stabilized after around 4 h. As we mentioned previously, research by Jarosz et al., 2014 [45] also showed that lactic acid produced by LAB could induce glucose suppression during the sourdough fermentation. Therefore, less glucose production in the LP-GM4-yeast group may be caused by lactic acid production (*L. plantarum-GM4*)uring fermentation. Furthermore, the glycemic index of the steamed bread obtained from the synergistic fermentation of LP-GM4 yeast was lower than that of the steamed bread fermented by only yeast starter (Figure 6). A research team also found that consumption of whole wheat bread produced by sourdough fermentation (with the addition of LAB as starters) provided higher SDS and RS and lower RDS and glycemic index values [40].

The above results indicate that the steamed bread produced by co-fermentation of *L. plantarum* and *S. cerevisiae* has low digestibility compared to steamed bread fermented with *S. cerevisiae* alone. We also found that co-fermentation of LP-GM4-yeast starter can delay the digestion rate of starch, which may aid in inhibiting the rapid rise in blood sugar, and is also beneficial in the prevention of diabetes and other related diseases. Following the above findings, we hypothesized that the slow starch digestion rate could be due to the co-fermentation of LP-GM4 yeast, as during fermentation LP-GM4 produces higher content of organic acids (lactic acid and acetic acid), which inhibits the activity of yeast and leads to less degradation of starch. Other researchers also presented similar findings: lactic acid produced by sourdough fermentation improved the RS ratio [49]. The type of flour, fermentation method, and temperature also significantly affect the RS value (*p* < 0.05). The *L. plantarum* fermentation results in the breakdown of long amylopectin into intermediate and short amylopectin, as well as the degradation of the amorphous area, and increases the relative crystallinity [50]. Exploring effects of lactic acid bacteria and yeast fermentation on rice starch molecules, Tu et al., 2021 concluded that sourdough microbial co-fermentation can induce the degradation and reorganization of starch molecules, thereby significantly affecting the digestibility of starch molecules [51].

#### 3.6.2. Protein Digestion

After centrifugation of the samples at different digestion stages, the results of the precipitates (Figure 7d), supernatants (Figure 7c) and undigested samples (Figure 7a,b) are shown in Figure 7 below. With increased digestion time, the number of polypeptides having a large molecular weight and subunits decreased. On the contrary, several proteins with small molecular weights increased in abundance, and the modifications were more obvious. Under the action of pepsin and trypsin, certain macromolecular proteins were degraded into small molecular polypeptides. In this study, when different undigested samples were treated, the molecular weight of the proteins in the steamed bread samples co-fermented with LP-GM4-yeast decreased. In contrast, the protein with low molecular weight increased, which improved the steamed bread quality with higher soluble proteins (Figure 8). Furthermore, with the addition of proteases, the protein degradation of macromolecules was considerably improved, especially the proteins with molecular weight of 30–40 kDa. Zotta et al. [52] reported that yeast has a negligible influence on protein degradation, while the effects of lactobacillus on the generation of medium and high-molecular-weight proteins are more apparent during sourdough fermentation. Almost all studies have indicated that sourdough fermentation increases in vitro digestibility of protein, which expresses the stability of protein hydrolysates and their digestion tolerance. Sourdough improves a variety of indices during prolonged fermentation, most prominently the protein content and nutritional quality based on the amino acid profile after hydrolysis and the nutritional index [53]. Zi-lin et al. [54] also used the means of dynamic simulation of in vitro digestion and concluded that sourdough fermentation improved the in vitro digestibility of steamed bread.

## 4. Conclusions

In this study, through the study of the physicochemical properties of three kinds of steamed bread with three different starters (blank, yeast and LP-GM4 yeast), as well as the nutritional substances starch and protein, it was found that the collaborative fermentation process of LP-GM4 and yeast resulted in a lower-pH environment in the dough and more organic acids. Further, compared to blank and yeast groups, the specific volume of steamed bread prepared by LP-GM4-yeast starter was increased approximately up to 175% and 32%, respectively. In addition, during in vitro digestion, it was found that the proportion of resistant starch in the cooperative fermentation of LP-GM4 and yeast increased, and the hydrolysis rate was lower during digestion, so the glycemic index decreased. Moreover, the large-molecular-weight protein of steamed bread was degraded, and more easily digestible soluble protein and precursor substances of synthetic flavor substances were produced, which improved the sensory score of steamed bread. Therefore, steamed buns produced by LP-GM4-yeast co-fermentation had healthier properties in terms of digestion.

## Figures and Tables

**Figure 1 foods-12-03333-f001:**
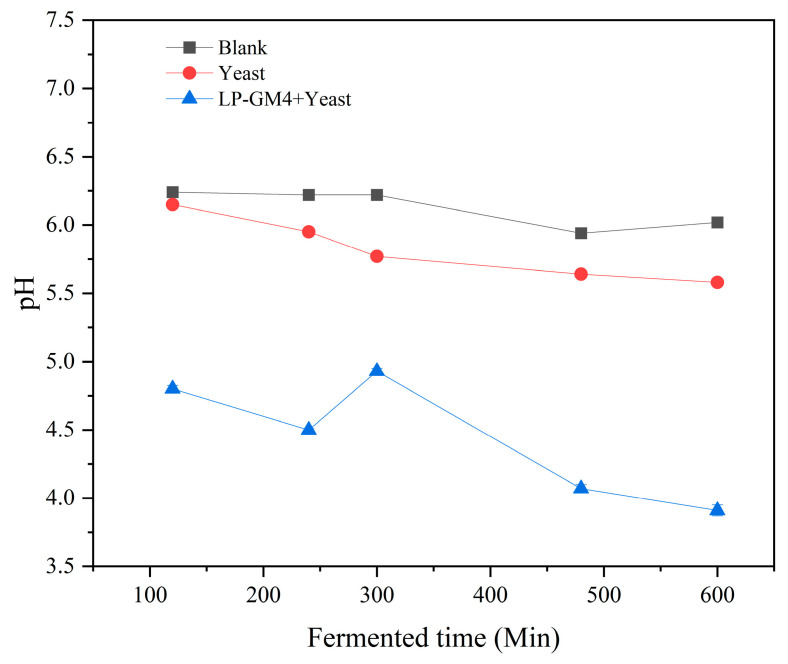
pH changes during fermentation with different starter cultures. The pH of steamed bread prepared with different starters, including yeast (*Saccharomyces cerevisiae*), LP-GM4-yeast (*Lactiplantibacillus plantarum* and yeast), and blank (control group) was measured after 2, 4, 6, 8 and 8 h. Values are presented as means ± SD (standard deviation).

**Figure 2 foods-12-03333-f002:**
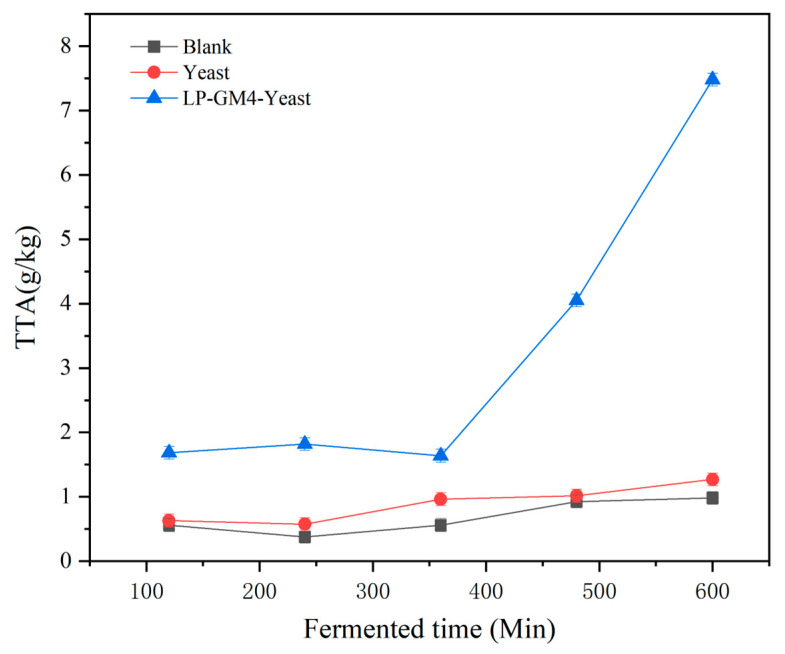
TTA changes during fermentation with different starter cultures. The titratable acidity of steamed bread prepared with different starters, including yeast (*Saccharomyces cerevisiae*), LP-GM4 yeast (a combination of *Lactiplantibacillus plantarum* and yeast), and blank (control group) was measured after 2, 4, 6, 8 and 10 h. Values are presented as means ± SD (standard deviation). Findings indicated that the LP-GM4-yeast group had the highest TTA rate after 10 h of fermentation.

**Figure 3 foods-12-03333-f003:**
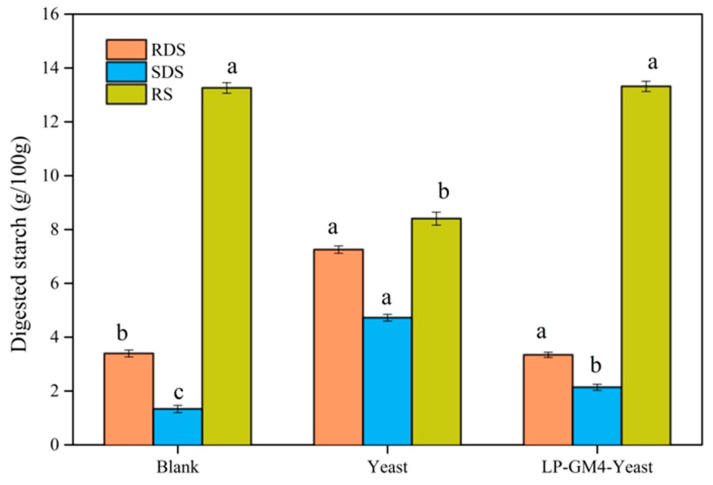
Comparison of in vitro digestion of various starches in different fermented steamed breads. The in vitro starch digestibility was analyzed using Chi et al.’s 2018 [31] technique. Rapidly digestible starch (RDS), slowly digestible starch (SDS), and resistant starch (RS) of steamed bread produced by yeast starter (*Saccharomyces cerevisiae*), LP-GM4-yeast starter (*Lactiplantibacillus plantarum* and yeast) and control group (blank) were measured. Values are presented as means ± SD (standard deviation). Results showed that compared to yeast alone (*Saccharomyces cerevisiae*) the proportion of RS of the LP-GM4-yeast group was significantly increased (>2 times). Different letters indicate significant differences.

**Figure 4 foods-12-03333-f004:**
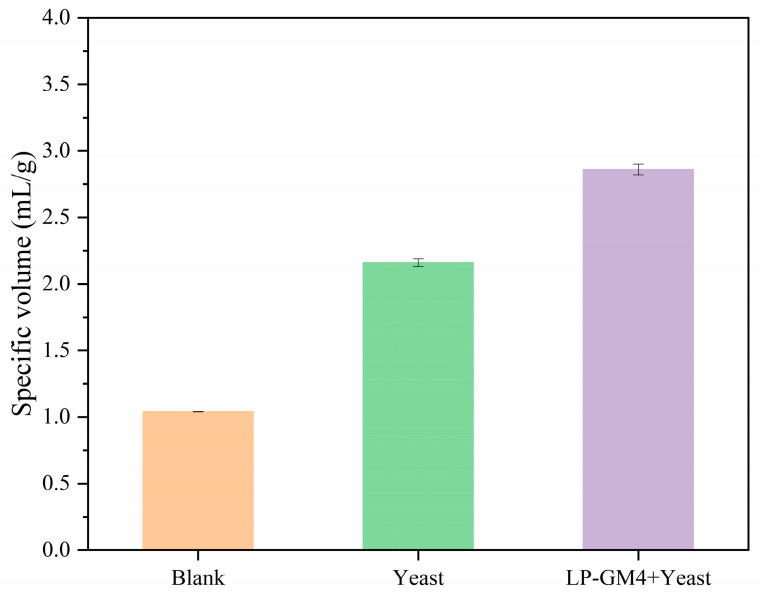
Specific volume of steamed bread fermented by different starter cultures. According to the GB/T 21118-2007 appendix [29], the millet seed displacement principle was used to measure the specific volume of steamed bread. The specific volume of the steamed bread produced by two types of starters (yeast (*Saccharomyces cerevisiae*), and LP-GM4 yeast (*Lactiplantibacillus plantarum* and yeast)) was determined. Blank indicates volume of the sample that did not undergo any fermentation. Unit of specific volume is mL/g. Values are presented as means ± SD (standard deviation). Results showed that compared to yeast alone, the specific volume of steamed bread fermented by a co-culture of LP-GM4-yeast was increased by about 32%.

**Figure 5 foods-12-03333-f005:**
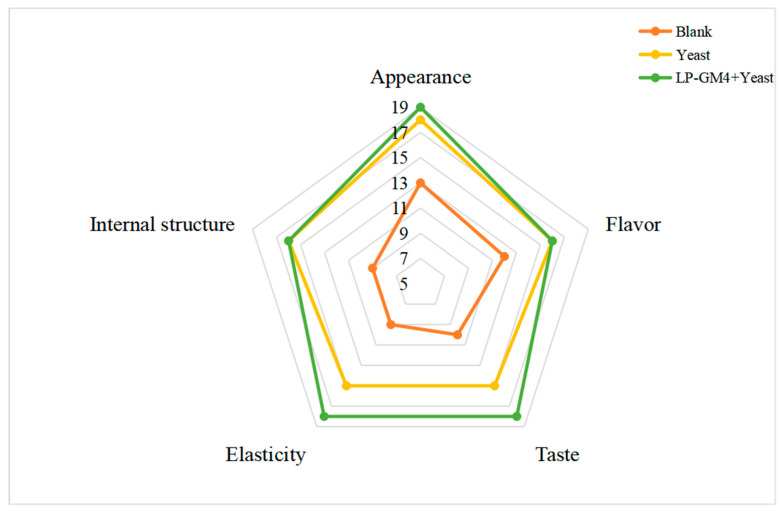
Sensory evaluation of different fermentation steamed bread. The appearance, internal structure, flavor, elasticity, and taste of the steamed bread produced with yeast alone and LP-GM4-yeast in combination and without starters (blank). Values are presented as means ± SD (standard deviation).

**Figure 6 foods-12-03333-f006:**
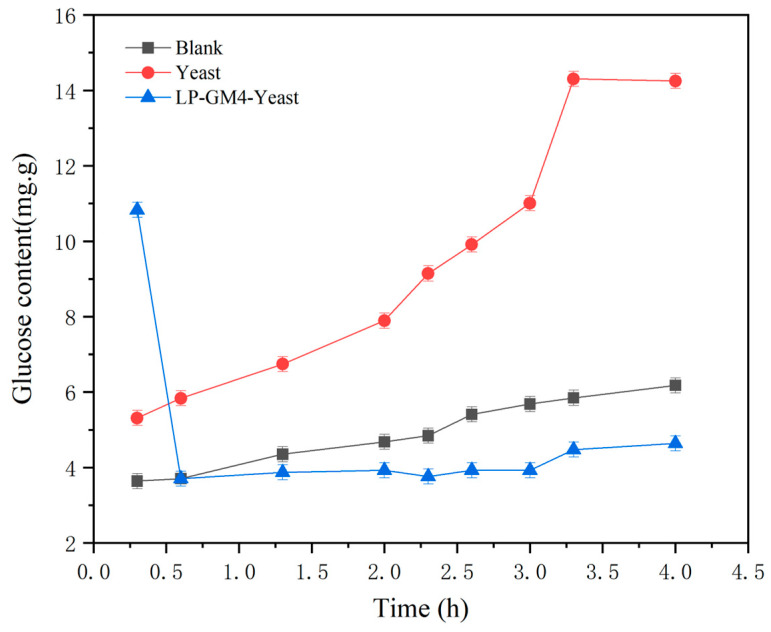
Changes in glycemic index during in vitro digestion of different fermented steamed bread. The glycemic index (glucose content) of the steamed bread fermented with yeast starter and the mixture of yeast and bacteria (LP-GM4-yeast starter) after 0.5, 1.5, 2, 2.5, 2.7, 3, 3.5 and 4 h was determined. Blank represents the control group. Values are presented as means ± SD (standard deviation). Results indicated that yeast steamed bread had the highest glucose content compared to the yeast and blank groups.

**Figure 7 foods-12-03333-f007:**
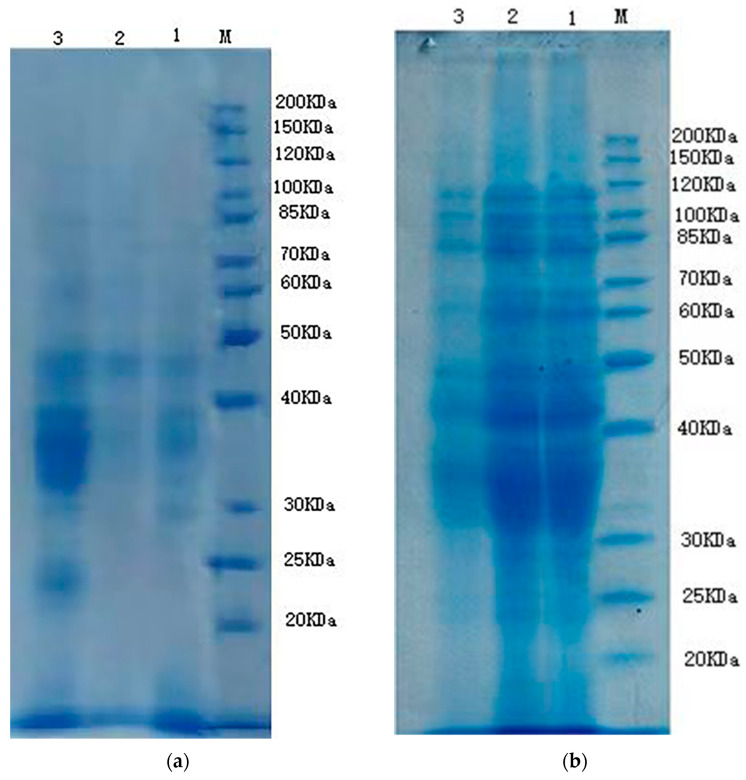
Protein digestion of steamed bread fermented with different starter cultures. (**a**) Water soluble protein; (**b**) Total protein; (**c**) digestion supernatant; (**d**) digestive solution precipitation. Protein digestion of the steamed bread fermented with yeast starter and the mixture of yeast and bacteria (LP-GM4-yeast starter) was determined using SDS-PAGE. M represents standard protein, 1, 2, and 3 represent undigested blank group steamed bread, yeast group steamed bread, and LP-GM4-yeast group steamed bread, respectively; Figure 8c,d represent the digestion supernatant and digestive solution precipitation during the protein digestion process, 1–9 represents blank group oral cavity, yeast group oral cavity, LP-GM4-yeast group oral cavity, blank group gastric, yeast group gastric, LP-GM4-yeast group gastric, blank group intestinal, yeast group intestinal, and LP-GM4-yeast group intestinal digestion.

**Figure 8 foods-12-03333-f008:**
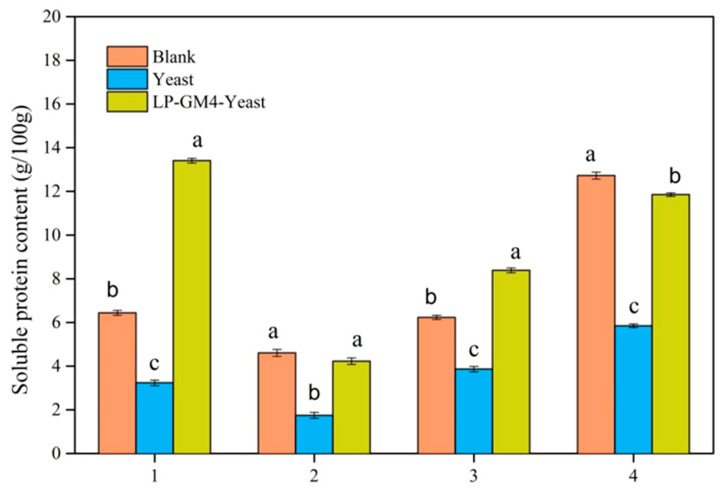
Soluble protein content of steamed bread fermented by different starter cultures. Concentration of soluble protein of blank, yeast and LP-GM4-yeast group steamed bread samples. Group 1 represents samples of undigested steamed bread; group 2 represents samples of oral-digested steamed bread; group 3 represents samples of gastric-digested steamed bread; group 4 represents samples of intestinal-digested steamed bread. Values are presented as means ± SD (standard deviation). Lowercase letters indicate significant differences (*p* < 0.05) among samples. Different letters indicate significant differences.

**Table 1 foods-12-03333-t001:** Composition of simulated digestive juice. SSF: simulated saliva fluid; SGF: simulated gastric fluid; SIF: simulated intestinal fluid. Unit of measurement was mmol/L. Values are presented as means ± SD (standard deviation).

	KCL	KH_2_PO_4_	NaHCO_3_	NaCL	MgCl_2_·6H_2_O	CaCl_2_·2H_2_O
SSF	15.1	3.7	13.6	-	0.15	1.5
SGF	6.9	0.9	25	47.2	0.12	0.15
SIF	6.8	0.8	85	38.4	0.33	0.6

**Table 2 foods-12-03333-t002:** Scoring criteria for sensory evaluation of steamed bread prepared with different starters, including yeast (*Saccharomyces cerevisiae*), LP-GM4-yeast (*Lactiplantibacillus plantarum* and yeast), and blank (control group).

Project	Scores	Scoring Criteria
Appearance	20	The shape of the bread is complete and the color is even milky white (15–20); the color of the steamed bread is white or slightly yellow and uniform (10–15); yellow or gray, uneven (5–10); surface shrinkage and collapse (0–5)
Flavor	30	Wheat flavor, moderate acid and alkali, pleasant taste (20–30); fragrance is light, taste slightly sour or alkaline (30 points); taste (10–20 points); taste sour (0–10)
Texture	20	Steamed bread has moderate viscosity and soft, easy to swallow (13–20); steamed bread is firm and does not stick to teeth (7–12); hard taste, hard to swallow (0–7)
Elasticity	10	Recovery faster (7–10); rebound slower (4–6); basic, no rebound (0–3)
Internal structure	20	The internal structure of the steamed bread is clear and the stomata size is uniform (13–20); the steamed bun is uniform and the stoma is smaller (7–12); stomata were hornet’s nest, no texture, and there is a lump (0–10)

**Table 3 foods-12-03333-t003:** The free amino acids produced during the digestion of steamed bread prepared with different starters (yeast (*Saccharomyces cerevisiae*), LP-GM4-yeast (*Lactiplantibacillus plantarum* and yeast and blank (control group)).

Amino Acid Type		Steamed Bread Type		
		Blank	Yeast	LP-GM4-Yeast
Essential Amino Acids	Thr	0.121	0.135	0.134
	Val	0.174	0.194	0.191
	Met ***	0.060	0.066	0.080
	Ile	0.167	0.184	0.182
	Leu ***	0.317	0.342	0.334
	Phe ***	0.228	0.246	0.234
	Lys ***	0.096	0.111	0.117
Semi-Essential Amino Acids	His	0.126	0.137	0.145
	Arg	0.157	0.170	0.171
Non-Essential Amino Acids	Ser	0.208	0.227	0.215
	Asp	0.182	0.205	0.211
	Glu	1.442	1.560	1.513
	Pro	0.529	0.552	0.526
	Gly	0.160	0.173	0.174
	Ala	0.138	0.155	0.176
	Cys	0.000	0.000	0.000
	Tyr	0.096	0.108	0.105
	TAA	4.202	4.565	4.507
	EAA	1.163	1.277	1.271
	NEAA	3.038	3.289	3.236

*** means *p* < 0.05.

## Data Availability

The data that support the findings of this study are available from the corresponding author upon reasonable request.

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
