# Peer review of "Effects of Co-Fermentation of Lactiplantibacillus plantarum and Saccharomyces cerevisiae on Digestive and Quality Properties of Steamed Bread"

_foods, 2023, doi:10.3390/foods12183333_

Round 1
Reviewer 1 Report
This manuscript presents interesting data that the quality of steamed bread was significantly improved by using Lactiplantibacillus plantarum. The information in the study should be of interest to many people who consume steamed bread. However, I have a few comments that the authors might consider.
Comment 1
There is no description about sensory evaluation in Material and Methods. Please add description.
Comment 2
The circles, squares, and triangles in Figure 1 and 2 is small and they are a little difficult to distinguish. I recommend increasing the size of them.
Comment 3
In lines 240-242, the author described that the specific volume of the bread obtained by fermentation of LP-GM4- yeast was increased compared to the blank bread by 134%. However, according to the sizes of the bars in Figure 4, the ratio 134% is probably different. Please check the data.
Reviewer 2 Report
The authors presented a study about the in vitro digestion of starch and protein in the sour-dough based steamed bread fermented by yeast alone or co-culture of LP-GM4- yeast. Because of the following reasons, this particular manuscript will not be considered further by "Foods".
1- I note the limited quality of English writing, which sometimes makes the sentences difficult to understand.
2-There is not enough novelty in the manuscript.
3- The results were not enough scientific.
Sincerely,
The authors presented a study about the in vitro digestion of starch and protein in the sour-dough based steamed bread fermented by yeast alone or co-culture of LP-GM4- yeast. Because of the following reasons, this particular manuscript will not be considered further by "Foods".
1- I note the limited quality of English writing, which sometimes makes the sentences difficult to understand.
2-There is not enough novelty in the manuscript.
3- The results were not enough scientific.
Sincerely,
Reviewer 3 Report
Effects of Co-fermentation of Lactiplantibacillus plantarum and Saccharomyces cerevisiae on Digestive and Quality Properties of Steamed Bread
Authors have selected good topic and conducted a valuable trial but there are some suggestions that may improve the quality of article.
In line 16 authors are claiming that 40 percent of Chinese are using steamed bread? Is it true? Do you have any reference?
Authors should mention statistical analysis in abstract at least in one line.
Give a conclusive line in the end of abstract.
Authors should elaborate the role of starter culture in introduction section.
Please use italic wording when writing the scientific name in the whole article.
Add importance, reasoning and aim of the study in end of the introduction section.
Please avoid to use abbreviation or at least mention one time in brackets.
In Figure 3 comparison is not properly discussed please elaborate more briefly and give proper comparison.
Sensory evaluation has not been mentioned in methodology section please add it.
Authors should properly explain the results along with their comparison, justification and reasoning in the discussion.
There are a lot of grammatical mistakes and language issues, please thoroughly revise the article.
No references are there from 2022 and 2023, authors are encourage to update references and add latest references.
Check references according to the journal format.
Extensive English improvement is required.
Reviewer 4 Report
1. Always use the Italic font to present the scientific name both in the title and text.
2. Abstract should revise for language.
3. Give the most important results and conclusions in the ending part of an abstract
4. Introduction should focus on how much work was completed on stream bred.
5. What is the importance of this work?
6. What are the problems associated with existed fermentation process?
7. Dont justify the usage of the method used to study in part of the introduction
8. The steam bread production method should be clear
9. In the materials and method, all the method's citations should add
10. Some of the sections in materials and methods need some elaboration, like in 2.8
11. All the images should convert to color including graphs
12. Discussion of all the parameters should improve
13. In Table 2, give the units for Aminiacids, I hope if we Present in the % of Amini acid increased from the blank may give good insight to the readers.
14. Table 1, foot note in inappropriate, use standard format
15. The procedure for the sensory analysis must be in materials and methods.
16. Table 3, also should come in materials and methods.
17. Electrophoresis pictures (figure 7) are not clear, must replace
18. Conclusions should be to the point based on the objective.
19. Concentrate on editorial issues throughout the document
1. Once, the manuscript should be revised for fine-tuning the language with the language expert.
Round 2
Reviewer 1 Report
The authors have responded appropriately to my comments.
I have no further comments.
Reviewer 3 Report
Authors have improved well and it can be accepted in present form
Language is much better but still there are some sentences in introduction and discussion section that needs more improvements.